# Water-Sensitive Urban Plan for Lima Metropolitan Area (Peru) Based on Changes in the Urban Landscape from 1990 to 2021

Andrea Cristina Ramirez Herrera [1,*], Sonja Bauer [2] and Victor Peña Guillen [3]

1    Institute of Spatial and Regional Planning (IREUS), University of Stuttgart, 70174 Stuttgart, Germany
2    Department of Electrical Engineering, Media and Computer Science, Ostbayerische Technische Hochschule Amberg-Weiden, 92224 Amberg, Germany
3    Department of Land Use Planning and Construction, Universidad Nacional Agraria La Molina, Lima 15012, Peru
*    Correspondence: andrea.herrera@ireus.uni-stuttgart.de

**Abstract:** Lima is the second-largest capital of the world located in a desert and already faces water scarcity. Here, more than 30% of the population is supplied by only 2.2% of the national water resources. The urbanization process has an informal nature and occurs at a very accelerated rate. These new settlements lack water infrastructure and access to other services. The objectives of this study are to quantify changes in the urban landscape of Lima Metropolitan Area from 1990 to 2021 to propose a water-sensitive urban plan by detecting changes, urbanization trends and identifying alternative water sources. The trend suggests a future constant increment of the urban areas, diversification of the landscape and more equally distributed land cover. Lima has more disconnected settlements and more complex shapes of urban patches nowadays. The landscape is also more mingled, but cracked. Overall, the trend is to become more disaggregated, demanding small and scattered water solutions. The WSUP includes the implementation of treatment plants in new multi-family buildings, hybrid desalination plants at the coast and parks with fog collectors on the hills. Additionally, these solutions will require the beneficiary community and the local authorities to work together in the planning and maintenance.

**Keywords:** urban changes; landscape metrics; Lima; water sensitive urban planning; water planning

## 1. Introduction

In the year 1950, only 746 million people in the world lived in an urban environment [1]. That had shifted rapidly by 1990, where 2.29 billion resided in built-up areas [2]. Latin American cities are the most populated areas in the world, after North America and Europe. By the year 2018, on average 80,7% of Latin American inhabitants lived in a city [1]. The increment of urban population between 1990 to 2020 was very rapid, the average annual rate being equal to 1.79%, and it is expected to follow this trend for the next decades [2]. Lima is regarded as a megacity due to its large population and accelerated growth. Every year, the increasing population in Lima Metropolitan Area (LMA) occupies new land, building informal settlements, which lack access to basic services [3] and add complexity to the landscape.

Urbanization also affects ecological dynamics risking the wellbeing of the environment, along with that of people who inhabit the cities [4]. At the intersection of natural resources and human activities, urban dwellers are consumers and also polluters of their environment [5]. Thus, environmental and socioeconomic dynamics modify not only the population distribution and behavior, but also ecological processes [6]. In this context, a megacity such as Lima struggles to cope with rapid urbanization, water scarcity and sustainable development. For that purpose, the water problem in Lima must be approached from three dimensions: the nature and environment, environmentally friendly economic

growth and social equity [7]. Planning a desertic megacity under these dimensions is challenging. Therefore, an important step is to analyze the changes in the urban landscape over the last decades to understand dynamics of that particular spatio–demographical context.

The current territorial and urban planning legislation in Peru provides the framework for the spatial organization of the activities in LMA (Law Nr. 31313, published on *El Peruano* on 15 July 2021). Nevertheless, these guidelines do not address neither the necessity nor the urgency of a water sensitive urban plan. In addition to that, local governments are autonomous and therefore, allowed to make planning decisions in a decentralized way according to the *Ley orgánica de municipalidades*. This is an advantage that would allow the current methodology to be applied locally and convert urban areas into water sensitive communities.

To accomplish this objective, the integrated methodology developed here combines the spatiotemporal analysis of LMA using landscape metrics [8] with the concept of water-sensitive urban planning. From the first part, a trend of urban growth and the quantification of spatial patterns were obtained. Furthermore, with the water-sensitive approach [9], solutions have been developed according to the emerging spatial context and to the availability of water resources at local scale.

The urbanization process in LMA has been rapidly shifting the configuration and composition of the urban landscape. These changes have not been investigated at the metropolitan scale for the period 1990–2021. In order to understand which processes and tendencies the urban landscape in LMA follows, landscape metrics (at class and landscape levels) are computed. These help to understand composition, configuration and distribution of patches [10], or as in this case urban units. Additionally, they have the potential for assessing and conceptualizing landscape patterns, functions and processes [11] to solve real-world problems. They can be applied to a variety of fields and dimensions to better understand processes and have been used to investigate urbanization [12–18] in different cities around the world. Nevertheless, the interpretation of landscape metrics for water-sensitive urban planning is a novel methodology that responds to the research questions regarding what we are quantifying urban changes for and how we can translate landscape metrics into tangible solutions to improve cities.

Water-sensitive urban design (WSUD) is a set of principles and practices to integrate the water cycle and urban planning. It follows three main principles [9] that should be understood as guidance and must be adapted to the local context. These principles are water supply strategy, provision of ecosystem services (EESS) and water-sensitive communities in cities [19]. By achieving a water-sensitive system, climate change effects are mitigated, and the heat island effect is reduced [20], which is a very important attainment for impervious centers. This concept and its application have been studied extensively, and in 2020 Esmail and Suleiman [21] analyzed 100 articles published since 2015. From the sample, the majority corresponded to studies located in Australia and the United States. They found a good number of publications in other continents, but none from South America.

The decision on what to propose varies, and it is applied to the local context, as well as the availability of resources. The most frequent procedures consist of qualitative methods. For instance, one research proposed rain collectors on the peri-urban zones of Aguascalientes, Mexico that recharges existing natural water reservoirs [22]. They also recommended the increment of green infrastructure for the intraurban localities based on the landscape characteristics and climatic conditions. The studied area is smaller than Lima in terms of surface and population and the proposed solutions will not work in a megacity that lacks rainfall episodes. In comparison, water-sensitive strategies were proposed in Dhaka, Bangladesh based on the spatial morphology and location of green and blue networks [23]. Dhaka is comparable to LMA with regard to increasing population and housing demand. This city also faces strong rural–urban migration, and therefore the water demand is substantially higher every year. For this reason, we believe that spatiotemporal analysis supports planification and anticipates problems. The trend of composition and

configuration of cities offers a large amount of information concerning suitable locations and sizes of the solutions to be constructed.

On the other hand, planification will not succeed if the communities do not shift to water-sensitive societies, where the population is actively involved in the process [24]. The impediments for further implementation are mostly related to policy frameworks, such as found in southern Australia [25]. The legislation does not make it mandatory to implement water-sensitive solutions. Added to that, they found that the lack of information about the maintenance tasks and costs plays a role against the adoption of water-sensitive strategies.

Investigating changes in the urban landscape is the starting point for analyzing other aspects of cities [26]. Landscape ecology provides a framework to measure land cover patterns and relate them to ecological processes. The current research explores relations between spatial patterns of the growing LMA, relating it to water supply. The results provide the foundations to propose location-based WSUD according to the local alternative water resource availability. The calculation of class and landscape metrics indicate how configured, aggregated, diverse and shaped the urban patches in LMA are. From the results, a trend was identified, thus providing a hint on how LMA will further develop in terms of urbanization.

The development of this methodology attempts to locate and propose water-sensitive solutions to shorten the water gap in megacities by applying an objective analysis of demand and supply. This tool can be used at different scales to translate the landscape metrics into tangible and quantifiable water-providing solutions. This paper presents a set of water-sensitive solutions adjusted to the different local contexts in LMA and their urbanization trend.

## 2. Materials and Methods

### 2.1. Study Area

Lima is the capital of Peru, and it is regarded a megacity due to its large population, more than 10 million people representing ca. 35% of the total Peruvian population live there [27]. This coastal city is located at the west side of the continent at the latitude 12.0464° S and longitude 77.0428° W. Lima is considered the second-biggest capital located in a desert (BWn in the Köppen climate classification) with mean annual precipitation equal to 9 mm in the metropolitan area [28]. The mean temperature is equal to 19 °C and it experiences high levels of humidity, reaching values up to 100%. Lima, a desertic mega-city, faces a water scarcity problem due to the climatic conditions, growing population and political instability. The water supply in LMA does not increase as fast as the population. Indeed, by 2014 the water deficit was equal to $-3.22$ m$^3$/s in winter and $-1.82$ m$^3$/s in summer [29]. The summer months in Lima coincide with the rainy season in the Andes, where the rivers of Lima are born. During these months, it is expected to have more discharge and less water deficit. Nevertheless, even in summer the amount of available water is not enough to cover the population demand. The situation worsened by 2019 and 2020, increasing the deficit to $-7.33$ m$^3$/s and $-7.88$ m$^3$/s, respectively [30]. This problem is accentuated by new inhabitants informally occupying areas that are not connected to the public network.

Lima is politically divided into Lima region, Lima province and the constitutional province of Callao. These last two make up LMA, the study area of this investigation, which is divided into five zones and 50 districts (Figure 1). It has an area equal to 2782.28 km$^2$. One international airport and one harbor are located here. The Callao harbor is one of the most important in Latin America, occupying the sixth place in the region and moving more than 2 million TEU [31].

Three watersheds conform the surface water reserve in LMA and represent only 2.2% of the national water resources [32]. The Chillon river passes through Lima North and Callao to finally flow into the Pacific Ocean (Figure 1). In 2021, the discharge of the Chillon river was equal to 14.8 m$^3$/s, which was 73% more than the historical mean (7.93 m$^3$/s) and 198.4% more than the mean in 2020 (4.96 m$^3$/s) [33]. The second basin is the Rimac. This

river had an increase of the mean discharge in 2021 (65.62 m³/s) that represents 73% and 98.2% more than the historical and 2020 mean, respectively. These discharge values were collected from the National Service of Meteorology and Hydrology of Peru (SENAMHI) and described by the National Institute of Statistics and Informatics.

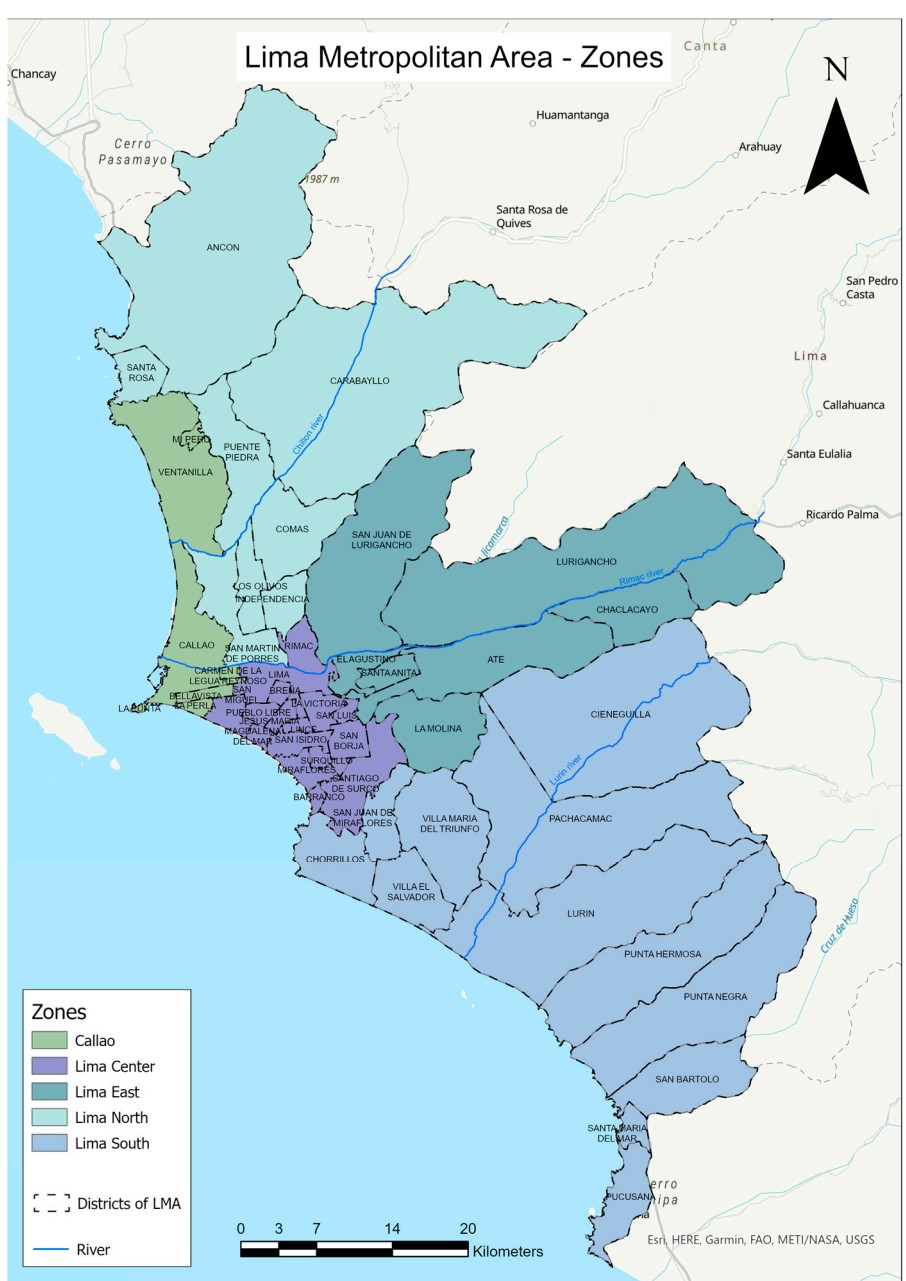

**Figure 1.** Lima Metropolitan Area. The territory is divided into five zones and 51 districts.

The potable water in LMA is administrated by SEDAPAL (*Servicio de Agua Potable y Alcantarillado de Lima*—Drinking Water and Sewage Service of Lima), an institution that produces clean water from surface (Rimac and Chillon rivers) and underground sources. The water purification process is ensured through physicochemical and bacteriological analyses, schedules and daily continuous monitoring of the water at the different stages, supported by a system that measures the behavior of variables in real time, such as pH, conductivity, turbidity and discharge [30]. Although the discharge of both rivers increased in 2021, the production of potable water decreased from 66,714,000 m³ in 2020 to 65,241,000 m³ in 2021 [33]. The third basin, and the smallest, is Lurin, located in the south of LMA. This

river also contributes to the production of fresh water for consumption but at a smaller scale. However, it is very relevant for the recharge of the aquifer and groundwater. The mean discharge of the river Lurin measured at the Pachacamac station is 3.95 m$^3$/s.

Lima also faces the effects of the El Niño–Southern Oscillation, a phenomenon that occurs irregularly and produces severe changes in climatic conditions. This causes the normal climate of Lima (arid) to become wetter. Recently, in 2017 El Niño Costero, a severe event, hit the coasts of Peru and Ecuador. It caused the humidity to increase, triggering heavy rains that led to overflows, floods and landslides in many Peruvian and Ecuadorian localities [34]. During this event, the discharge of the Rimac river increased, causing the service provider company SEDAPAL to close the intake points to avoid capturing mud. This meant that 27 districts did not have drinking water for days. The Peruvian coast shows great vulnerability to drastic climatic variations, such as extreme episodes of rain and high temperatures associated with El Niño and extreme climatic events occurring in the Andes [35].

Although the increment of precipitation favors the recharge of the aquifer, the negative impacts outweigh the benefits. The occurrence of El Niño accelerates glacier retreat, clogs the reservoirs, destroys water and sanitation infrastructure, increases the probability of forest fires, increases the occurrence of diseases such as cholera [36]. These impacts are directly related to water security and availability in Lima.

In LMA, 80% of the water resources come from the Rímac river basin (the remaining 20% come from the Chillón and Lurín basins). This is not sufficient to satisfy the capital's demand of 700 million m$^3$ per year, projected by the Drinking Water and Sewage Service of Lima. In fact, as of 2015, the people of Lima demanded 28% more than the actual available drinking water [30]. By 2040, the gap will rise to 65%, according to SEDAPAL projections.

Many climate change simulations (optimistic and pessimistic scenarios) indicate an increase of temperature of 1–3 °C in the next decades, as well as a reduction of precipitation, in South America [37]. This will lead to accelerating glacier meltdown, thus affecting the Andean region, where the three rivers of LMA are born. The direct consequence will be a shortage of superficial water availability. Added to this, when vegetation shifts to urban land cover, it causes an alteration of the water cycle [38,39], affecting the growing water demand of the increasing population.

In LMA, 96–97% of the buildings have access to water through the public network [40]. This percentage includes connections inside and outside the house, as well as public-use pylons. Nevertheless, this value is an average that includes official settlements, very well-supplied districts, as well as districts that have little to no access to the resource. But most importantly, this average excludes the informal new urban settlements on the periphery of the city. The other way of supplying drinking water is through tanker trucks. For instance, the districts of Lima North, Ancon (46%) and Santa Rosa (57%); Lima South, Pucusana (57%) and Punta Hermosa (65%); and Lima East, Cieneguilla (45%) are mostly supplied by trucks and other sources [41].

## 2.2. Methodology

Landscape metrics has been extensively studied and applied in different fields. Nevertheless, most of the studies end with calculation and interpretation [11,12,42–46]. In this case, our starting point is the landscape metrics calculated for LMA for the years 1990, 2008 and 2021. From here, we developed a methodology that combines the interpretation of the metrics with the quantification of urban development to propose a water sensitive urban plan for the metropolitan area of Lima.

The classified images were produced from Landsat-5 (Path 007 row 068, path 007 row 069. Dates: 9 February 1990 and 14 April 1996) and Sentinel-2 (T18LTM, T18LTN. Date: 20 April 2021). They were downloaded from the USGV Earth Explorer and from the Copernicus Open Access Hub, respectively. The images were layer-stacked (multiband images), mosaiced and clipped to the extent of LMA using the software ERDAS [47]. Afterwards, a supervised classification using the nearest neighbor method was performed.

For the first iterations, the four main classes were divided into 12 to 16 subclasses and signatures were collected accordingly. This decision was made due to the quality of the input images and the wavelength differences within one class. The result was recoded to display only four land uses that include the urban areas, the class of interest. The class "Bare-land" includes hills, bare-land in the city and at the coast, arid territories and non-vegetated areas. The land use "urban" displays well-consolidated built-up areas, slums, single-family and industrial buildings. Agricultural fields, green urban areas and swamps were categorized as "vegetation". Roads, streets and highways, small water bodies such as lakes, treatment plants with a visible water surface and other small structures not fitting any other class were grouped in "other". Lima was simplified into four classes in order to focus on the analysis of the urbanization process.

Following this, a majority filter was applied to generalize and reduce single-pixel misclassification. Afterwards, masks were established, especially for the big bare land areas on the outskirts to reclassify these pixels. Finally, small, misclassified areas were manually reclassified in ArcGIS PRO [48]. The classified images were prepared to be used as input in the FRAGSTATS software [8].

The calculation of class and landscape metrics indicates how configured, aggregated, diverse and shaped the urban patches in LMA area are and were used to indicate the nature of the water-sensitive strategies to be applied. Hence, seven metrics at class and seven at landscape level were computed to highlight the direction of LMA urban development. The first group includes class area (CA), the surface of landscape covered by a patch type; percentage of landscape (PLAND), the proportion of urban areas in relation to the total LMA; mean shape index (SHAPE_MN), the relation between the actual perimeter of a patch and the hypothetical perimeter of it, if it would be maximally compacted; mean fractal dimension index (FRACTAL_MN), which describes the complexity of the urbanized centers, and the alternative area-weighted fractal dimension index (FRAC_AM), which shows the complexity weighted to the patch area; proportion of like adjacencies (PLADJ), which indicates the frequency how often patches of different classes and urban patches are next to each other using the double-count method; and aggregation index (AI), ranging from 0 to 1 and indicating how aggregated or disaggregated urban settlements are. To complement the analysis of urban areas, metrics at landscape level were computed, such as mean contiguity index (CONTIG_MN), to asses connectedness of cell in patches and the alternative area weighted contiguity index (CONTIG_AM) multiplied by the proportional abundance of classes; division (DIVISION), which provides the probability that two random cells are not collated in the same class; contagion (CONTAG), indicating the probability that two random cells belong to the same class; Shannon's diversity index (SHDI), which displays the diversity of classes and their abundance; Shannon's evenness index (SHEI), which measures the proportion of classes and their distribution; and aggregation index.

Using multitemporal analysis, the urban expansion trend was identified, thus providing a hint on how LMA will further develop in terms of urbanization. These processes change the demand and supply of clean water in different zones of the metropolitan area, where the availability of the resource varies from zone to zone (and district to district). The results were matched with the availability of alternative water resources to propose measurements for producing water from AWSS. Following the three principles proposed by Wong and Brown [19], the water supply strategy, as well as the provision of additional ecosystem services and the proposal of water–sensitive communities were adjusted to the local context. The summary of the proposed methodology for the integration of a water sensitive urban plan for LMA is shown in Figure 2.

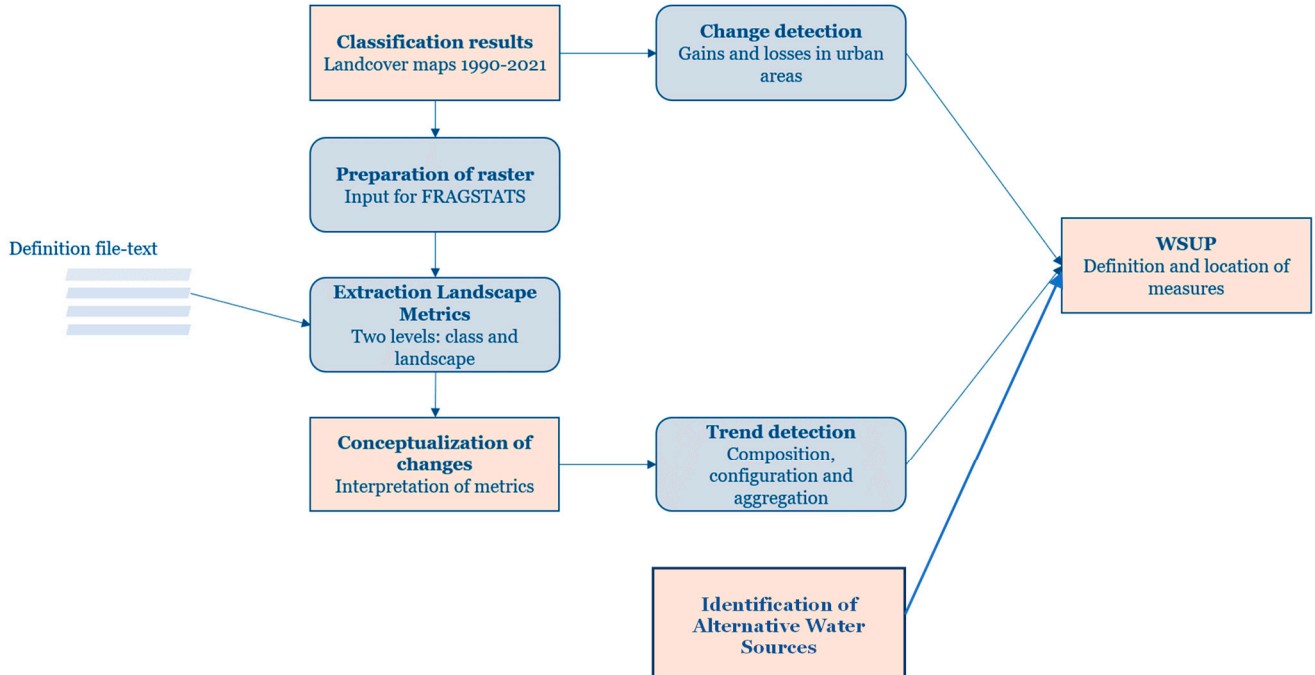

**Figure 2.** Methodology for the integration of a water-sensitive urban plan for Lima Metropolitan Area.

## 3. Results and Discussion

### 3.1. Change Detection: Gain and Losses in Urban Areas

The results showed that all classifications (Figure 3) were acceptable. The overall accuracies ranged from 80.1% to 95.5%. The urban settlements increased by 106.9% between 1990 and 2021, with a yearly rate of 3.5%. Two administrative units located in the metropolitan area of Lisbon (Portugal) experienced a similar growth [42], increasing by 91.1% in built-up areas between 1990 and 2000. Similarly, this phenomenon was observed in Tshwane, Guangdong and Mumbai [16,43,44]. The urban areas in the city of Tshwane increased by 109.1% between 1985 and 2015 and expanded to non-populated territories. Additionally, the urban sprawl corresponds to population growth, which also showed high growing rates. In Guangdong, the urban patches expanded, surrounding the consolidated areas and forming many small nuclei. Similarly, new settlements in Lima occupy former bare lands and vegetation areas in the outskirts of the city center. By 1990, 2126.4 km$^2$ were identified as bare land (Table 1). This value representing mostly the desolated territories on the hills declined by more than 330 km$^2$ in 2021. The impervious area more than doubled from 1990 to 2021. According to the current quantification, vegetation was reduced only by 80 km$^2$. Since this class includes agricultural fields as well as green areas in general, it is likely that the increase in green urban areas has compensated for the change in natural vegetation cover. A more meticulous analysis with focus on the change in vegetation in LMA would be necessary to correctly interpret this value. The area of the class "other" shifted from 1.5 km$^2$ (1990) to 19.7 km$^2$ (2021) due to the implementation of treatment plants and other civil engineering structures with superficial water surfaces.

In overall, the urban areas of the five LMA zones shifted during the study period. Nonetheless, Lima Center (LC) was the only zone displaying a negative sprawl (−14.25%). Figure 4 displays the urban sprawl per period and zone. It is noticeable that the areas further west of LMA (Callao and Center) were almost completely covered by urban settlements by 1990. Although Lima North (LN), South (LS) and East expanded considerably and even more than Lima Center, it is detectable that there are other dominant covers in these zones. The land cover of these areas may potentially shift in the coming years. LN more than doubled the urban areas (+105.45%). Following the trend, LS almost quadrupled the area

of settlements (+256.16). Callao and Lima East also experienced a significantly high urban sprawl, corresponding to +66.19% and +76.98%, respectively.

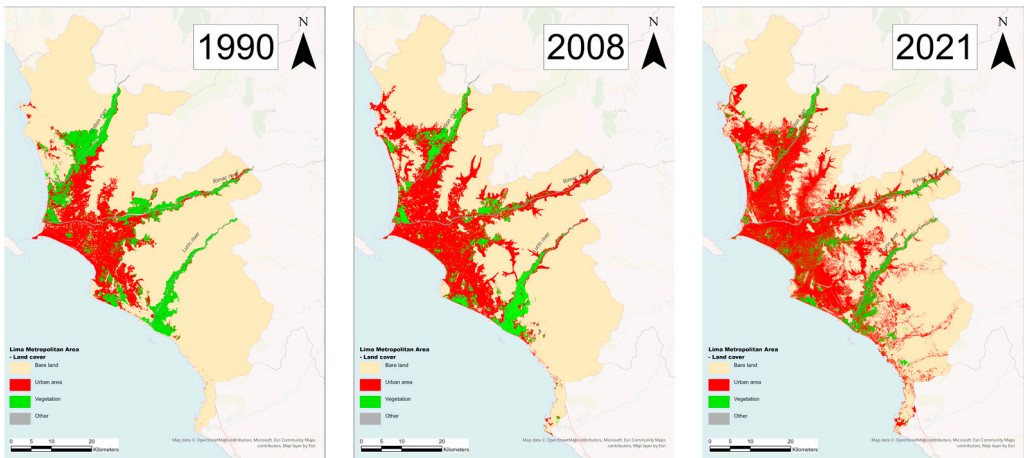

**Figure 3.** Land cover changes in Lima Metropolitan Area for the years 1990, 2008 and 2021.

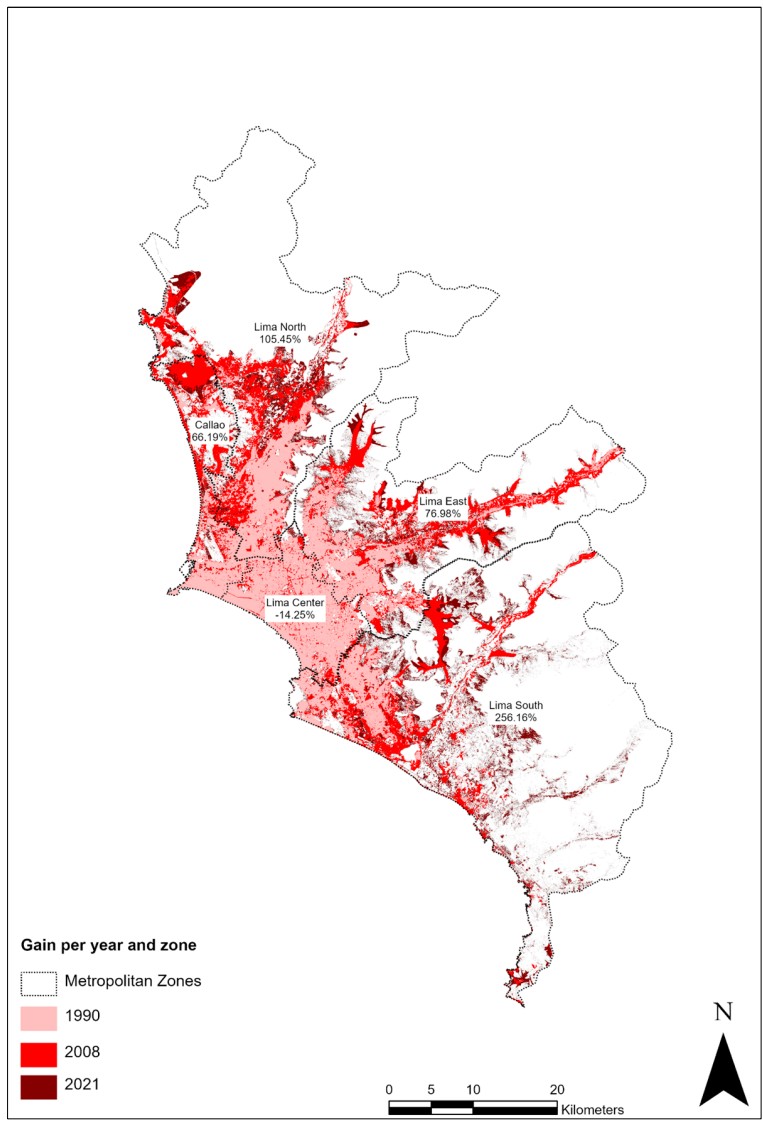

**Figure 4.** Urban area changes 1990, 2008 and 2021.

**Table 1.** Landscape metrics at class level for Lima Metropolitan Area.

| YEAR | CLASS | CA (km$^2$) | PLAND | SHAPE_MN | FRAC_MN | FRAC_AM | PLADJ | AI |
|------|-------|-------------|-------|----------|---------|---------|-------|----|
| 1990 | Bare land | 2126.39 | 76.426 | 1.204 | 1.041 | 1.178 | 98.745 | 98.81 |
|      | Urban | 381.03 | 13.695 | 1.237 | 1.043 | 1.322 | 92.064 | 92.206 |
|      | Vegetation | 273.36 | 9.825 | 1.188 | 1.038 | 1.223 | 91.695 | 91.862 |
|      | Other | 1.5 | 0.054 | 1.661 | 1.078 | 1.179 | 77.488 | 79.441 |
| 2008 | Bare land | 1939.18 | 69.84 | 1.185 | 1.039 | 1.154 | 98.667 | 98.734 |
|      | Urban | 627.16 | 22.613 | 1.246 | 1.044 | 1.335 | 93.434 | 93.546 |
|      | Vegetation | 194.88 | 7.053 | 1.192 | 1.041 | 1.208 | 89.105 | 89.297 |
|      | Other | 12.69 | 0.495 | 1.224 | 1.046 | 1.1 | 64.21 | 64.736 |
| 2021 | Bare land | 1787.96 | 64.263 | 1.181 | 1.047 | 1.385 | 95.016 | 95.038 |
|      | Urban | 788.37 | 28.336 | 1.183 | 1.045 | 1.472 | 85.605 | 85.635 |
|      | Vegetation | 186.21 | 6.693 | 1.15 | 1.042 | 1.272 | 81.463 | 81.523 |
|      | Other | 19.73 | 0.709 | 1.137 | 1.04 | 1.188 | 58.83 | 58.963 |

*3.2. Trend Detection: Interpretation of Landscape Metrics*

In 1990, more than 75% of the LMA was desolated, built-up areas covered 13.7%, while vegetation only 9.8% of the total extension (Table 1). The distribution of the values changed dramatically by 2021. The urbanized zones had more than doubled in 31 years and enclosed more than 28% of the total LMA. Whereas PLAND for bare lands and vegetation reduced by ca. 15% and 30%, respectively. A similar result was reported by Paudel and Yuan [45] while assessing landscape change in the Twin Cities Metropolitan Area (USA). The urban cover almost duplicated the percentage of landscape in the same time span (20% in 1975 and 36% in 2006). In LMA, human settlements occupied the bare zones at the foothills of the Andes. These areas are characterized by very steep slopes, which make the supply of water facilities, among other services, even more difficult. Bare land still represents a high proportion of the landscape and a potential expansion area for urbanization.

The SHAPE_MN of urban areas differs for the studied period by 0.054 (Table 1). Hence, the complexity of this class slightly reduced over time. Nevertheless, since the shape index was bigger than one, urban patches have had high geometric complexity since 1990. Analogous findings were reported by Magidi and Ahmed [12], the SHAPE_MN for the city of Tshwane was greater than one and attributed to the clustering of new small patches. In Lima, new unplanned urban units appear scattered on the outskirts of the metropolitan area. These new units decreased in complexity over time and suggest the need for small adaptable water-sensitive solutions.

The net difference of FRAC_MN for urban (+0.002) was positive and slightly higher in 2021 than in 1990. The distribution of patches can be understood as being marginally more fragmented by the end of the studied period. The FRAC_AM is another way to measure shape complexity. The FRAC_AM indicate that the difference of complexity of urban patches is small but positive and constant for all the periods. The increase of FRAC suggests irregularity in the pattern of urban settlements. Additionally, this parameter denotes the informality of the new settlements.

The value of PLADJ for the built-up areas decreased between the years 1990 and 2021. This value indicates a 7% reduction of the compactness, pointing out the appearance of single-urban units. In the same manner, the urban settlements grew disaggregated in LMA. These results are also supported by the aggregation Index, which decreased by 6.6 for the urban class.

In addition to the metrics at class level, metrics at the landscape level were calculated for the same periods and years (Table 2). These metrics analyze the composition and configuration of LMA as a whole. The CONTIG_MN decreased from 1990 (0.3) to 2021 (0.1), indicating loss of connectivity in the landscape. Likewise, CONTIG_AM showed the same behavior. CONTAG decreased over time from CONTAG$_{1990}$ = 69.7 to CONTAG$_{2021}$ = 57.2. As a deduction, the LMA landscape is characterized by being more interspersed in 2021 and

by showing a trend toward the homogenization of landscape patch classes. The AI value at a landscape level also shrank from 97.2 (1990) to 91.2 (2021), suggesting, as well as the AI at the class level, that the landscape grew disaggregated. Equivalently, DIVISION at a landscape level was equal to 0.8 in 1990 and increased to 0.9 over 31 years. Therefore, it can be interpreted as the LMA landscape splintering. The increment by 2021 indicates that more patches of classes are single cells than in 1990. The SHDI reveals the number of present classes and their abundance. This index increased in LMA during the studied period from 0.7 to 0.9. By 2021, more land covers were to be found in one area unit. Correspondingly, SHEI presents the diversity of classes and the distribution of these. The SHEI augmented over time, advocating an increase of the proportion of land cover and a better distribution.

**Table 2.** Landscape metrics at landscape level.

| YEAR | CONTIG_MN | CONTIG_AM | CONTAG | DIVISION | SHDI | SHEI | AI |
|------|-----------|-----------|--------|----------|------|------|-----|
| 1990 | 0.2934 | 0.9678 | 69.6952 | 0.7676 | 0.7098 | 0.512 | 97.2123 |
| 2008 | 0.2854 | 0.9625 | 65.784 | 0.8543 | 0.8001 | 0.5772 | 96.7274 |
| 2021 | 0.1179 | 0.9077 | 57.2195 | 0.8699 | 0.8576 | 0.6186 | 91.2133 |

Overall, the LMA landscape was more diverse in 2021 (SHDI$_{1990-2021}$ = +0.2), more disaggregated (CONTAG, DIVISION, CONTIG) and the distribution of patches became more regular (SHEI$_{1990-2021}$ = +0.1) than in 1990. The urban areas increased, doubling the initial total area as well as the number of isolated–disaggregated urban settlements. These new urban establishments have a complex shape, although their complexity reduced over time (SHAPE) and they are more fragmented (FRAC).

LMA is an urbanized territory, and the built-up areas constantly put other land coverages at risk. These values indicate that urban growth occurs in a very disorderly manner. Added to that, the high values of diversity and distribution suggest high potential for future land cover changes. Additionally, the scattered distribution of the new settlements represents a difficulty for the implementation of infrastructure to provide basic services in the new neighborhoods in the traditional way. The little connectivity in the landscape, as well as the high diversity of land cover make it difficult to plan major water infrastructure. Additionally, the formal infrastructure can be only implemented in households holding a property title [49], which most of the new urban units lack.

*3.3. Identification of Alternative Water Sources: Water Sensitive Urban Planning*

In order to better propose a solution, it is mandatory to identify alternative water sources available in the study area. The water supply strategy for Lima would exclude surface water from the three main basins and underground water, which currently provide the population with this resource and are already saturated.

The first alternative water-supply source (AWSS) is wastewater. After its use, wastewater in LMA goes to the existent aerobic, anaerobic and facultative ponds, activated sludge plants, artificial wetlands and trickling filters [50]. These plants are insufficient for producing high volumes of recycled water. Therefore, the first-identified alternative water supply source is wastewater, specifically greywater coming from households connected to the network. Greywater comes from domestic and industrial use, excluding water from toilets and urinals. Greywater can be treated easily to be used for irrigation or flushing toilets. With an additional step, greywater can also be utilized for washing clothes. According to Alaziz and Al-Saqer [51], greywater constitutes ca. 35% of total domestic water demand. Greywater reuse systems can reduce more than 50% of hot and cold water consumption when installed in multistory houses [52]. This system would be more appropriate in Lima Center, well consolidated urban zones with negative urban sprawl that mostly shift from one-family houses to multifamily houses. The greywater recycling solutions can be also implemented as artificial wetlands and be integrated in the landscape, providing additional ecosystem services [53]. Nevertheless, due to the risk of microbial and viral contamina-

tion [54], in LMA the recycled greywater will essentially be used for flushing toilets and for cautiously irrigating green urban areas.

A second AWSS is advection fogs, coming from the Pacific Ocean and going to the inversion zones above 400 masl. The LMA's hills have great potential for collecting water from advection fog seasonally. These hills are found in Lima North, South and East. Fog collectors operate in many parts of the world, and in South America principally in desertic areas in Ecuador, Chile and Peru [55]. In Peru, the biggest fog collection project was implemented in Mejia, Arequipa during 1995 and 1999. In LMA, the implementation of fog collectors has been made at very small scales (individual fog collectors) and in an empirical manner [56]. Nevertheless, fog collectors represent an alternative source that can shorten the water gap as proven in the Canary Islands, Spain [57]. The collectors there produce between 11 to 68 L/day at the different chosen locations. The fog collectors in LMA should be integrated in a cultural park that can be designed at a small scale. These parks can be located at high altitude areas, where the settlements are disaggregated, and the landscape divided.

Due to the proximity to the sea, the third AWSS is the salty water of the Pacific Ocean. The districts located in the north coast are the ones experiencing great urban expansion and population increment. Desalination plants in Australia came to supply a third of the population's demand and have been extensively studied to improve them in terms of energy and costs [58]. Desalination plants are seen as an expensive, severe and extreme solution. Politicians see the high cost of implementation and maintenance as a problem, while the scientific community focus on the carbon emissions of individual plants [59]. Nevertheless, research in this field has advanced, and these counterarguments are being dismantled. The energy needed to desalinate 1 $m^3$ has been significantly reduced and is nowadays equal to 3 $kW/m^3$ [60]. This demand can be met with renewable energy, more specifically to LMA, solar panels, biogas or tidal energy. In addition, to maintain the efficiency of the plants and reduce the emissions, plants must provide water locally and shorten transport distances [59]. In this way, the use of batteries is reduced, as well as production costs.

## 4. Water Sensitive Urban Plan for LMA

As analyzed in the last section, the tendency in LMA is to grow disaggregated, scattered, forming complex shapes and discontinued units. These new informal neighborhoods are characterized by being decentralized and auto-organized. For this reason, the water structures in the zones with positive sprawl must be designed at a small scale and scattered in location. They must provide for households locally and be of simple use and maintenance. These new structures must output clean water from alternative water sources and offer additional EESS. They have to be located in-between the urban patches to reduce costs in transportation and distribution.

Lima Center behaves differently than the other zones and showed a negative urban sprawl. By 1990, LC was already well-consolidated in terms of urbanization; the urban settlements had stopped expanding horizontally. In consequence, water-providing infrastructure must be included in the design of new multistory buildings. It can be said, due to the obvious real estate development, that this territory behaves as a compact city [61]. The mean number of stories in La Victoria, Lima, Rimac and Breña increased by +27.5% from 2017 to 2020 [62]. Likewise, other districts in LC showed vertical development, and the mean number of building stories augmented between 6-9.4%. The urban landscape change is visible, especially along the Via Expresa, one of the main roads in LMA. In the last decades, many minor and low buildings have been replaced as a consequence of the real-estate boom. One example of that occurrence is the new 33-story building LUX constructed after 2015 in Lince (Figure 5).

Currently, legislation supports the sustainable construction of new housing. The supreme decree approving the technical code for sustainable construction DECRETO SUPREMO N° 015-2015-VIVIENDA provides guidelines for the construction of housing

with high energy and water efficiency. The specifications related to water are as follows: (1) buildings have to be handed over to their owner with sanitary devices that include technology for saving water, (2) all new buildings must be delivered to their owner with sanitary installations for water-treated domestic waste, and (3) new buildings have to plant xerophytic vegetation in their gardens. Nevertheless, the application of these specifications is not legally binding. The certifications LEED (Leadership in Energy and Environmental Design—USA), VERDE (*Valoración de Eficiencia de Referencia de Edificios*—Spain) and BREEAM (Building Research Establishment Environmental Assessment Method—United Kingdom) are granted to the buildings fulfilling the requirements to ensure environmentally friendly energy and water infrastructure. For instance, the buildings of the Hotel Westin and the National Peruvian Bank (*Banco de la Nación*) located in LC are LEED certified.

Due to the water situation in LC (higher supply of water through the public network in comparison to other zones), the change in the composition (trend to replace low with high buildings) and the configuration (same category buildings/new urban settlements are not aggregated) of the landscape, we propose the following solutions for Lima Center: (1) Water supply strategy: greywater treatment plants. (2) Provision of EESS: treated water will be used for flushing toilets, watering green areas (cautiously) and maintenance of cultural open spaces with xerophytic vegetation. (3) Water-sensitive communities: the planning, design, implementation and maintenance of the solution is decided and agreed for the beneficiary population. The principles to be followed as in WSUP and the measures are the result of analyzing changes in the LC urban landscape add up to changes in the population, changes in verticality and availability of fresh water.

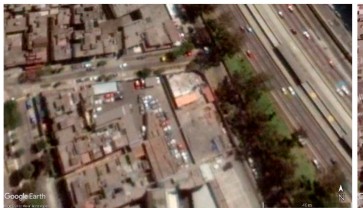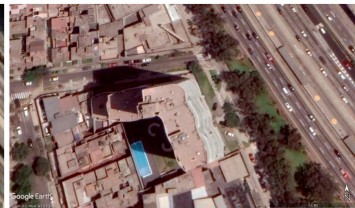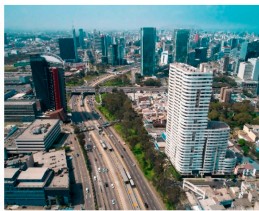

**Figure 5.** Replacing buildings—Building "LUX" in Lima Center. Satellite image from 2015 (**left**) and 2022 (**center**). Source: Google Earth, Maxar Technologies 2022. Photo of the building LUX in Lima (**right**) and other high buildings in the background. Source: https://edificiolux.com.pe/ (accessed on 5 November 2022).

The treatment of greywater must be included in the design of a building at the planning stage, before construction (Figure 6). Greywater can be separated from blackwater by installing a dual-plumbing system. The collected greywater from sinks and washing machines travels through the pipes and meets the treatment tank. In here, the filtration process takes place, and the water is separated in two: the reusable water (that travels with help of a pump back to the house and irrigation systems for gardens) and the water to sewer. The tanks are located under the gardens to save extra space or aboveground, depending on the availability and soil types. Small treatment plants can reuse wastewater at a building/house level for watering gardens and flushing toilets, saving the resource for other consumptions. Reusing wastewater not only contributes to the water gap problem, but also saves money and benefits the environment [63].

If the housing is already built, a predesigned treatment plant can be added. There are many producers, and the plants vary in size and efficiency. However, they follow the same principles: pretreatment, biological process, disinfection and filtration. The reuse of wastewater using this system is safe and ensures public and ecosystem health [63]. The advantage of these small plants is that they have volumes above 3000 L with capacity to clean the wastewater of a family of up to five members [64]. The tanks can also be implemented in parallel to satisfy the demand of more people. Another benefit is that they can be installed in already built-up areas. The disadvantages are that these tanks take up a large amount of space and the implementation is relatively expensive. This can be

counteracted by granting subsidies for the installation of small treatment plants. Planning at the time of construction also counterbalances the disadvantages. At this stage, the sale price of the new apartments can include the cost of installation. The treated water then contributes to the development of the community and offers added benefits [44], which is important in developing cities.

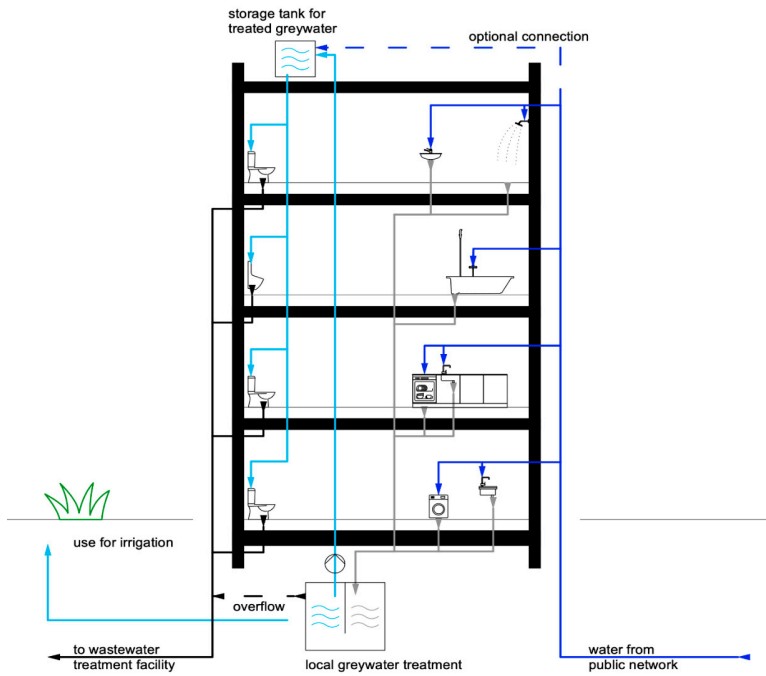

**Figure 6.** Diagram of the hydrologic cycle in a building with greywater treatment. Source: author.

For the South, East, North and Callao areas, with a positive sprawl and the appearance of informal settlements, the water supply strategy not only considers greywater/wastewater treatment, but also salty water and advection fogs. In these zones, there is also an interest in increasing the urban green area per inhabitant in the well-consolidated urban communities [65]. The following measure will be focused on Lima North and South, since the sprawl in these zones was the highest within the studied period.

In these areas, the implementation of additional greywater plants at a building or communal level will alleviate the water gap. In addition, it will increase the quality of life by implementing more green spaces necessary for health and well-being [66]. The greywater treatment plants can be similar to the ones proposed for LC. According to the Ley Orgánica de Municipalidades Ley N° 27972, the municipalities are granted political, economic and administrative autonomy. For that reason, they could implement regulations at local scale to mandate that the construction of new buildings include water recycling systems, as well as technologies to increase energy efficiency.

Due to its proximity to the northern coast, the water desalination plants would be located in the districts of Santa Rosa and Ancon (Figure 7) located in LN. In both districts, the public network supplies only 20 to 30% of the total amount of houses [67]. Currently, desalination plants with hybrid systems and batteries produce and transport 1 m$^3$ of potable water at a cost of around 2 EUR/m$^3$ [68], which is much less than what is paid for tanker-truck water in Lima. In 2017, 1 m$^3$ from the public network in LMA cost EUR 0.8, while people without access to the net paid EUR 4 for water from trucks [69]. The design of the desalination parks provides additional EESS, such as aesthetic and cultural, regulating and supporting services. In order to meet that purpose, the community should be involved in the planification and design of the plant and even integrate it into a cultural park (Figure 8). Since the water situation in LMA is extreme, the desalinated water can be recycled after the first use to further contribute to the water cycle. It has been found that evapotranspiration

rate to rainfall on land is equal to 75% [70], thus benefiting the hydrological conditions in the zone. Likewise, in the case of the southern LMA, the water desalination plants would be located in the districts of Villa el Salvador, Punta Hermosa and San Bartolo (Figure 7).

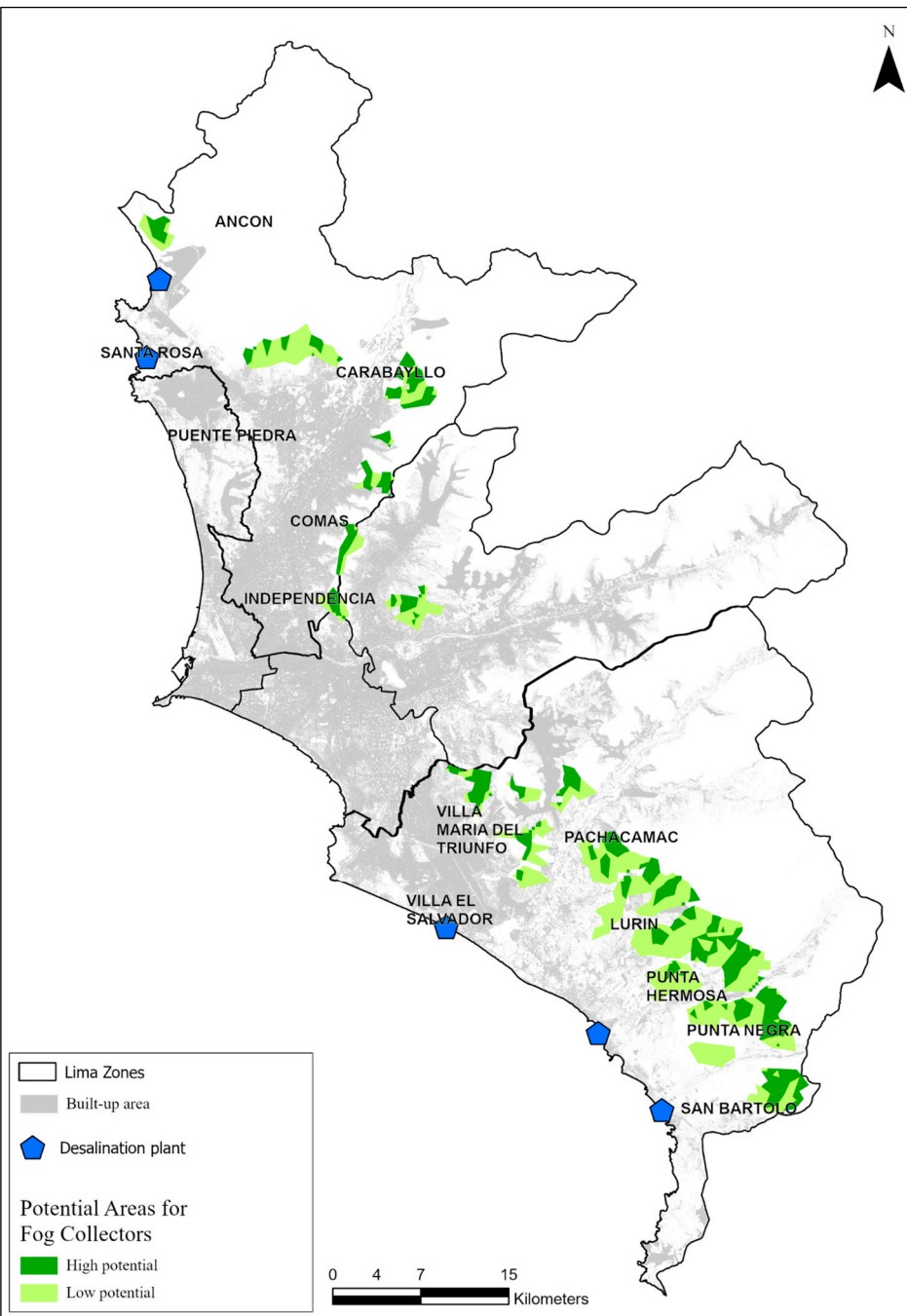

**Figure 7.** Location of the proposed solutions as part of the WSUP for Lima North and South. The map shows the built-up area by 2021 and the proposed location of desalination and fog collector parks in fast-growing districts.

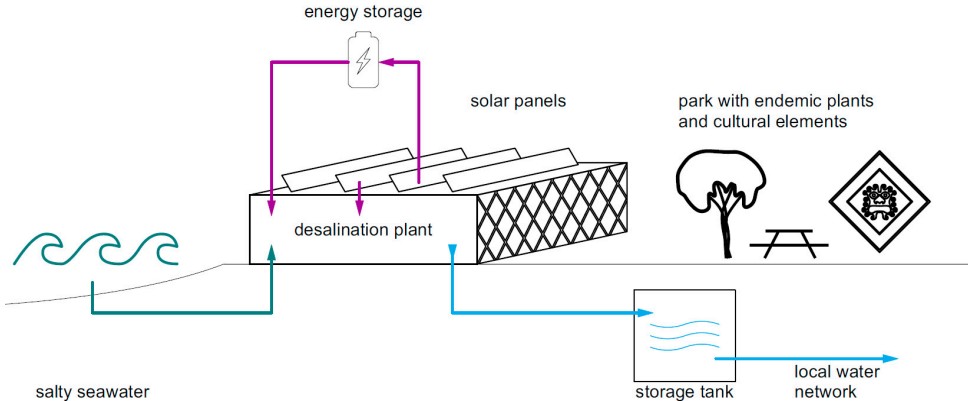

**Figure 8.** Integration of desalination plant and cultural park. Source: author.

Furthermore, another solution will be the implementation of fog collectors on the top of the hills. The northern hills are characterized by a very rugged relief, and ravines or dry torrents. The two most relevant hills in Lima North are Carabayllo and Ancon [71]. The first one is characterized for its plant cover stratified in herbs, shrubs and cacti. They have a high biodiversity and endemic species. Similarly, the Ancon hills have a very dense seasonal vegetation cover towards the Pasamayo highway and towards the interior of ravines. The hills are fundamentally upholstered by tillandsias and cacti, which make up xerophytic thickets whose dynamics and structure deserve special attention. The other hills located in the northern area are close to urban centers, for which their access is relatively simple; however, their steep and rocky nature and the social situation of each neighboring population pose limitations to individual access [71], but at the same time potential for water sensitive communities.

Fog collectors catch condensed water that passes into pipes when it returns to its liquid form (Figure 9). They represent an alternative non-conventional water source. In Lima, the higher potential for fog collectors is located at altitudes above 500 masl and faces west to the Pacific Ocean [72]. The ocean's continental advection fogs occur in the hills of LMA due to its proximity to the Pacific. Water condensation increases with altitude until reaching the thermal inversion zone (in Peru between 750 and 1800 masl). The collected water is then stored in a tank that must be located close to the structure. In a fog collector study in the city of Asir [73], it was found that the amount of collected water depends on meteorological conditions. The majority of water volume collected took place when the temperature was between 5 °C and 10 °C, the relative humidity >90% and the wind speed 4 m/s. Additionally, fog collectors have to be monitored monthly to ensure good water quality. The pipes and storage tanks cannot be exposed to sunlight, high temperatures or dust. These collectors should be also integrated in a park that will add value by providing other EESS to the community. Abualhamayel and Gandhidasan [73] calculated that one fog collector 20 m wide and 2 m high captured 2.06 L/m$^2$/day during a 72-day period including optimal condition days, rainy days and days with fog absent. The mean volume collected was equal to 82 L/day, which almost covers the water necessity (100 L/day) of one person according to the WHO. Nevertheless, the mean consumption in Lima equals 250 L/day/person [69]. The proposed location of fog collectors (Figure 7) will provide an alternative water resource to improve life quality in the zone, while reducing the price to be paid for covering the water demand.

In 2019, the Supreme Decree Establishing the Regional Conservation Area of the Lomas de Lima System Nº011-2019-MINAM was approved in order to ensure the conservation of these hills and to protect them against illegal invasion. This decree protects an extension of 10,962.14 Ha of hills located in the districts of Ancon, Carabayllo, Independencia, Rimac, San Juan de Lurigancho, La Molina and Villa Maria del Triunfo. The first-mentioned three districts are located in LN and the last in LS. Villa Maria del Triunfo has shown an accelerated growth in area and in population [40] and faces water-related problems.

These hills were selected and included as potential areas in the calculation for placing the fog collector parks. Hence, the design and implementation of these open spaces must be in accordance with the supreme decree N°011-2019-MINAM and be approved by the competent authorities.

In summary, the proposed water strategy, provision of EESS and water-sensitive communities in these two zones would benefit 13 districts. These measures would ease the water problem and shorten the supply–demand gap. Apart from producing clean water, a provisioning ecosystem service, the production of this must contribute to the creation of new extra ones that benefit human welfare. The Millennium Ecosystem Assessment (2005) made it clear that human wellbeing is directly linked to the natural environment through the conceptualization of EESS. Thus, the water supply strategy is designed in a way that additional EESS can be provided at the same time.

The community's values and aspirations should be considered while planning the city. For this purpose, local governments must listen to and work together with the population to encourage maintenance of the measures and long-term success. The local scale is key in addressing water supply in informal and new settlements [74]. Moreover, a sense of community is already present in the new urban settlements and must be exploited for the benefit of the population.

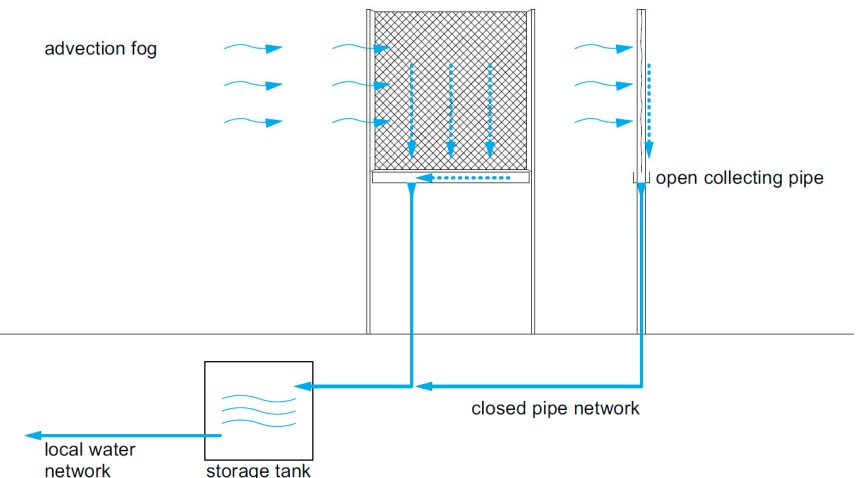

**Figure 9.** Scheme of a fog collector. Source: author.

## 5. Conclusions

Worldwide, the rapid urban growth of non-planned cities has negative consequences in the population wellbeing. Therefore, a proper analysis of changes and processes is the first step to understanding what phenomena are taking place in the territory. In this way, it is possible to identify the trend to be followed in the coming years and to design an urban plan accordingly. The chosen methodology has not been applied to proposing a water-sensitive urban plan before; however, the results indicate that it is an efficient way to locate and analyze changes in the landscape to propose planning measures. The results showed the trend of Lima of growing more disaggregated, divided, disconnected and expanding into bare land and the outskirts of the city. In most LMA zones, the new urban settlements are characterized by being made up of small single-built-up units that appear according to the needs of the population. However, these needs do not respond to any planning and have an informal and disorderly character. Lima Center did not follow the trend. This zone is the oldest urbanized environment, and instead of showing a large appearance of new houses, the phenomenon of densification is taking place. The change from single-family homes to multi-family buildings represents an opportunity to change the design approach. This fact can be taken as a good possibility to integrate water-sensitive strategies and gradually convert Lima into a water-sensitive city.

Having considered the alternative water sources in Lima, there is great potential to execute projects in the new neighborhoods. The advantage of having these sources scattered throughout the territory is an opportunity that offers the possibility to implement many small solutions. In this way, new urban areas will have locally produced clean water to cover a proportion of the total demand. The proposed technologies will not completely solve the water problem, but they provide information for the planning oriented toward shortening the supply–demand gap. In Lima, as well as in other cities in developing countries, changing the city model is not a luxury, but a necessity to covering the basic needs of the population. Nevertheless, this change would not only be covering basic needs, but also making Lima a city more resilient to climate change.

The beneficiary communities would refine the design details, especially those related to the provision of additional ecosystem services. The social factor is mandatory for ensuring the further functioning of the technologies and the success of the projects. Water-sensitive communities must be constituted and be involved in the process of planning, implementation and maintenance. Nevertheless, these solutions would not work without the support and commitment of local authorities, which is a major challenge in South American countries, where economic development does not go hand in hand with environmental planning. Thus, the success of water-sensitive strategies depends on the establishment of new policies. For instance, local governments can implement ordinances to force the inclusion of water-sensitive solutions in the urban environment or to protect existing initiatives.

**Author Contributions:** Methodology, investigation, analysis, writing-original draft preparation, A.C.R.H.; review and supervision, S.B. and V.P.G. All authors have read and agreed to the published version of the manuscript.

**Funding:** this research received no external funding.

**Institutional Review Board Statement:** Not applicable.

**Informed Consent Statement:** Not applicable.

**Data Availability Statement:** Not applicable.

**Conflicts of Interest:** The authors declare no conflict of interest.

## Abbreviations

The following abbreviations are used in this manuscript:

| | |
|---|---|
| LMA | Lima Metropolitan Area |
| WSUD | Water-sensitive urban design |
| EESS | Ecosystem services |
| CA | Class area |
| PLAND | Percentage of landscape |
| SHAPE_MN | Mean shape index |
| FRACTAL_MN | Mean fractal dimension index |
| FRACTAL_AM | Area weighted fractal dimension index |
| PLADJ | Proportion of like adjacencies |
| AI | Aggregation index |
| CONTIG_MN | Mean contiguity index |
| CONTIG_AM | Area weighted contiguity index |
| DIVISION | Division index |
| CONTAG | Contagion |
| SHDI | Shannon's diversity index |
| SHEI | Shannon's evenness index |
| AWSS | Alternative water-source supply |
| LC | Lima Center |
| LN | Lima North |
| LS | Lima South |

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
