# Peer review of "Water-Sensitive Urban Plan for Lima Metropolitan Area (Peru) Based on Changes in the Urban Landscape from 1990 to 2021"

_land, doi:10.3390/land11122261_

Round 1

Reviewer 1 Report

The manuscript titled " Water Sensitive Urban Plan for Lima Metropolitan Area (Peru) based on changes in the urban landscape from 1990 to 2021 " intends to quantify changes in the urban landscape of Lima Metropolitan Area from 1990 to 2021 to propose a water sensitive urban plan by detecting changes, urbanization trends and identifying alternative water sources. 

The research is original; it could be characterized as novel and in my opinion important to the field, it also has an almost appropriate structure, and the language has been used well. In the meanwhile, the manuscript has a quite nice extent (about 6,770 words) and it is quite comprehensive. The tables (2) and figures (9) make the paper reflect well to the reader. For this reason, paper has a "diversity look", not only tables, not only numbers, not only words.

The title, I think, is all right. The abstract reflects well the findings of this study, and it was the appropriate length. The introduction is effective, clear, and well organized but it wasn’t introduced and put into perspective what research is negotiating. Moreover, it does not contain a clear formulation and description of the research problem. Please insert a clear description and justification of the problem the article deals with. Your literature research should be critical and more informed, rather than listing previous research. This section requires significant improvement.

The most important problem here is that you do not develop the Literature review, a very important element of every research work that is done and it is the section where it is also used in the Discussion, comparing the literature with the results you found.

For the Methodology chapter, the research conduct has been tested in several areas of the world, with comparable results and will probably be tested in others. Appropriate references to the methodology included in the already published bibliography but you can put more references, from all over the world. Do not forget, the journal “Land” is international.

The results section is good. The argument flows and is reinforced through the justification of the way elements are interpreted. But the same does not apply to the Discussion and Conclusion. It is advised to revise the Discussion and Conclusion. Both sections should be consistent in terms of Proposal, Problem statement, Results, and of course, future work. Your conclusion section does not do justice to your work. Make your key contributions, arguments, and findings clearer. You must refer to the literature and previous studies in your discussion section.

Please revise the manuscript and include more references which already exist in the bibliography. I would be much more satisfied if the number of references was slightly higher (about 15 - 20 references) and I would appreciate it if it also included data from all the world Asia, America, Europe, or Australia. In this way it is documented that a method that is tested in a place with its own characteristics can be implemented in other places around the world.

More discussion is needed, comparing the results of this work related to attributes with those of other studies. I believe that the conclusions section or discussion should also include the main limitations of this study and incorporate possible policy implications. I think something more should be said about practical implications.

Also use the appropriate research manuscript sections: Introduction, Materials and Methods, Results, Discussion and Conclusions, as journal wants (https://www.mdpi.com/journal/land/instructions).

Please use the appropriate Figure captions and the appropriate Table captions & footers (https://www.mdpi.com/files/word-templates/land-template.dot). 

Please fill in the subchapters accordingly as: Author Contributions, Funding, Institutional Review Board Statement, Informed Consent Statement, Data Availability Statement, Acknowledgments and Conflicts of Interest, according to the instructions of the International Journal Sustainability [see: Instructions for Authors / Manuscript Preparation/ Back Matter - (https://www.mdpi.com/journal/land/instructions#submission or https://www.mdpi.com/files/word-templates/land-template.dot)].

Author Response

Dear sir or madam,

thank you for your comments. You can find a specific response attached to this message.

Kind regards,

Andrea Cristina Ramirez Herrera

Author Response

(The authors gave the same response as above.)

Reviewer 3 Report

Dear Authors,

the topic covered is very interesting and the proposed solutions seem like an excellent idea, the article is clear and well written. However, the article would need some adjustments, because in some parts it is sometimes long and not well balanced. For example, whenever the 4 groups emerge, their formalization or description should be better structured. The Introduction should be a little expanded, perhaps delving deeper into the topic, and partly also the conclusions, for example adding limits and future perspectives of the work. The methodology should also be described better, giving it a little more space. In any case, after some small changes the article can be published. Please, find below some specific suggestions that I hope will help.

·       Acronyms spelled out the first time

·       row 67-71: Is the water deficit higher in winter than in summer?

·       Lines 192-199: the 4 identified areas could be better schematized

·       Line 223: “gain” and not “Gain”

·       Line 225-227: One decimal is enough

·       Lines 239-240: you could better interpret and investigate those aspects

·       Paragraph 4.1: they should make a similar speech for each of the 4 identified areas

·       Line 308: table 10 maybe should be table 1

·       Line 350: “water” not “Water”

·       Line 429-430: single spacing, Figures and not Fig

·       Line 435: “I proposed…” Maybe it's better “we proposed…”

·       line 495: check “plants 28” the footnote is missing

·       Line 578-579: single spacing, Figures and not Fig

·       Check the references according the PDPI rules

Author Response

(The authors gave the same response as above.)

Round 2

Reviewer 2 Report

No comments in this round.